# Equine Penile Squamous Cell Carcinomas as a Model for Human Disease: A Preliminary Investigation on Tumor Immune Microenvironment

**DOI:** 10.3390/cells9112364

**Published:** 2020-10-27

**Authors:** Ilaria Porcellato, Samanta Mecocci, Luca Mechelli, Katia Cappelli, Chiara Brachelente, Marco Pepe, Margherita Orlandi, Rodolfo Gialletti, Benedetta Passeri, Angelo Ferrari, Paola Modesto, Alessandro Ghelardi, Elisabetta Razzuoli

**Affiliations:** 1Department of Veterinary Medicine, University of Perugia, 06126 Perugia, Italy; ilariaporcellatodvm@gmail.com (I.P.); samanta.mecocci@studenti.unipg.it (S.M.); chiara.brachelente@unipg.it (C.B.); marco.pepe@unipg.it (M.P.); margherita.orlandi94@gmail.com (M.O.); rodolfo.gialletti@unipg.it (R.G.); 2Centro di Ricerca sul Cavallo Sportivo, University of Perugia, 06126 Perugia, Italy; 3Department of Veterinary Science, University of Parma, 43126 Parma, Italy; benedetta.passeri@unipr.it; 4Zooprophylactic Institute of Piemonte, Liguria and Valle d’Aosta, 16129 Genova, Italy; angelo.ferrari@izsto.it (A.F.); paola.modesto@izsto.it (P.M.); elisabetta.razzuoli@izsto.it (E.R.); 5Azienda Usl Toscana Nord-Ovest, UOC Ostetricia e Ginecologia, Ospedale Apuane, 54100 Massa, Italy

**Keywords:** papillomavirus, penile cancer, horse, tumor immune microenvironment, carcinoma, squamous cell, models, animal

## Abstract

Penile squamous cell carcinomas (SCCs) are common tumors in older horses, with poor prognosis mostly due to local invasion and recurrence. These tumors are thought to be mainly caused by *Equus caballus* papillomavirus type 2 (EcPV-2). The aim of this study is to characterize the tumor immune environment (TIME) in equine penile tumors. Equine penile epithelial tumors (17 epSCCs; 2 carcinomas in situ, CIS; 1 papilloma, P) were retrospectively selected; immune infiltrate was assessed by histology and immunohistochemistry; RT-qPCR tested the expression of selected chemokines and EcPV-2 DNA and RNA. The results confirmed EcPV-2-L1 DNA in 18/20 (90%) samples. *L1* expression was instead retrieved in 13/20 cases (65%). The samples showed an increased infiltration of CD3^+^lymphocytes, macrophages (MAC387; IBA1), plasma cells (MUM1), and FoxP3^+^lymphocytes in the intra/peritumoral stroma when compared to extratumoral tissues (*p* < 0.05). Only MAC387^+^neutrophils were increased in EcPV-2^high^ viral load samples (*p* < 0.05). *IL12*/*p35* was differentially expressed in EcPV^high^ and EcPV^low^ groups (*p* = 0.007). A significant decrease of *IFNG* and *IL2* expression was highlighted in *TGFB1*-positive samples (*p* < 0.05). IBA1 and CD20 were intratumorally increased in cases where IL-10 was expressed (*p* < 0.005). EpSCCs may represent a good spontaneous model for the human counterpart. Further prospective studies are needed in order to confirm these preliminary results.

## 1. Introduction

In humans, penile carcinomas, of which 95% are reported to be squamous cell carcinomas (SCCs), are a relatively common health issue, particularly in developing countries, where the estimated incidence is reported to reach up to 3.2 cases per 100,000 men, in the face of an incidence rate usually <1 reported in the United States and Europe [1]. The incidence of penile cancer is lower than cervical cancer, which is known to be mostly caused by high risk human papillomaviruses (hrHPVs) infection [2]. HPV DNA has been found in a lower percentage of penile SCCs (30-50%) [3], when compared to cervical carcinomas, but the pathogenesis of HPV-induced cervical and penile tumors is thought to be similar. Hence, the difference in terms of HPV-positivity could be due to a lower susceptibility of the penis to malignant transformation and other factors that may concur in the pathogenesis of penile SCCs [1,4,5]. Even though the diagnostic and clinical approach to SCCs is the same for hrHPV^-^ and hrHPV^+^ penile tumors, patients with hrHPV^+^ are reported to show better disease-specific survival than the hrHPV^-^ ones [6,7]. Papillomaviruses (PVs) are small, non-enveloped viruses, with a double-stranded genomic circular DNA and an icosahedral capsid [8]. Their pathogenetic role has been recognized in humans and animals in benign lesions as well as in malignant tumors [8,9,10,11,12,13]. Nevertheless, PVs can also be associated with persistent asymptomatic infections [8,14]. These viruses are highly host-specific and they have a strong tropism for stratifying epithelia, both on skin and mucosal sites, and, in humans, are classified into low-risk HPVs and high-risk HPVs [15]. The latter have been demonstrated to be involved in the pathogenesis of cervical, anogenital, head, and neck cancer (mostly squamous cell carcinomas, SCCs) [16].

In horses, growing evidence has been pointing at *Equus caballus* papillomavirus type 2 (EcPV-2) as the causal agent of equine penile SCCs (epSCCs), which represent the most common tumor of the external male genitalia [17]. These tumors often arise in horses of about 20 years, and they develop from plaques and papillomas, which are considered to be the precursor lesions [18]. The prognosis is poor, due to late presentation, local invasiveness and recurrence, and deterioration of the animal’s health [19]. Recently, epSCCs have been suggested as a potential spontaneous model for human penile cancer, given the numerous histological and immunophenotypical similarities of the tumors in the two species [20].

The host’s immune system is known to play a pivotal role in many tumors, including SCCs and PV-induced cancer [7,8,21,22]. In humans, the tumor immune microenvironment (TIME) of penile SCCs has been recently investigated by Chu and colleagues and was found to be predictive of nodal metastasis and prognosis [23], confirming its importance in tumor development and progression. Moreover, the results from previous studies on human penile SCCs indicate FoxP3 as an independent predictor of recurrence [24]; the expression of PD-L1 by tumor cells, a CD163^+^ macrophage infiltration, a non-classical HLA class I upregulation, and a low stromal CD8^+^ T cell infiltration have also been associated with nodal metastasis [7].

In equine penile SCCs, the characterization of immune infiltrate has been explored in occasional studies, but information on their TIME is still minimal [18,25].

The aim of this study is to better characterize the TIME of equine penile SCCs, by describing the immune infiltrate and expression of the main chemokines that are involved in tumor-associated inflammation, also assessing the correlations of these factors with the pathological findings and the EcPV-2 status.

## 2. Materials and Methods

### 2.1. Case Selection

Cases were retrospectively selected from the archives of the Department of Veterinary Medicine of the University of Perugia and the Department of Veterinary Science of the University of Parma on the following inclusion criteria:histological diagnosis of equine papilloma, carcinoma in situ, and squamous cell carcinoma, assessed by two boarded pathologists (IP, CB), as per recently suggested diagnostic criteria [17];confirmed penile localization of the lesions; and,availability of >0.5 cm^2^ of FFPE tumor tissue evaluated on section.

When available, non-lesional skin without neoplastic tissue from surgical margins was selected, together with the neoplastic tissue, for histological and immunohistochemical comparison.

### 2.2. Histopathological and Immunohistochemical Characterization of Tumors and of the Immune Infiltrate

After the histological diagnosis was confirmed, differentiation, mitotic count [26], vascular invasion, ulceration, intratumoral necrosis, and quantity of tumor stroma were evaluated for each case. A semi-quantitative evaluation (grade 0–3) of inflammation, and of different inflammatory cell populations were assessed in a double-blinded fashion by two pathologists (IP and MO). Whether concordance among the two pathologists was not present, a third pathologist (CB) re-evaluated the slides.

For immunohistochemistry, FFPE samples were cut into 5 μm sections, mounted on poly-l-lysine coated slides, dewaxed, and rehydrated. Immunolabeling was obtained with standard protocols on serial sections, with antibodies anti-CD3, FoxP3, CD20, MUM1, MAC387, IBA1, and Ki-67, as summarized in Table 1. The immunolabeling was revealed with 3-amino-9-ethilcarbazole (Abcam); Mayer’s hematoxylin was applied as a counterstain. Table 1 reports the expected reaction pattern. Negative controls were run by omitting the primary antibody incubating sections with TBS. Control-tissue was also tested with antibody isotype (only for monoclonal antibodies). Positive cells were manually counted in intra/peritumoral (areas within the tumor area and at the periphery of the tumor, but still in contact with tumor cells) [27] and extratumoral areas (clean surgical margins from the same animal), avoiding areas of necrosis and/or near ulceration. Ki-67 index was evaluated on 1000 cells by one operator (MO), while using ImageJ cell counter and expressed as a percentage, as previously reported [28].

### 2.3. DNA Extraction and Evaluation of EcPV-2 Viral Load

Five sections (5 um) of FFPE samples were used for DNA extraction, which was performed while using AllPrep DNA FFPE Kit (Qiagen, Venlo, Netherlands) in accordance to the manufacturer’s instructions. EcPV2-L1 DNA was tested while using specific primers set and probe previously described [26] and primer for Beta-2-Microglobulin (*B2M*) gene to test the DNA amplifiability. All of the primers are reported in Table 2. Real-Time PCR amplification was performed on a CFX96™ Real-Time System (Bio-Rad, Milan, Italy); 5 μL of template were added to 25 μL PCR mixture at final concentration of 1×master mix (Canvax, Cordoba, Spain), 200 nM of probe, 100 nM of each primer with the following thermal profile: 95 °C for 10 min, then 40 cycles of 95 °C for 15 s and 60 °C for 60 s. An estimation of the viral load was given, as previously described [27], on the basis of the cycle quantification (Cq) number at which was detected EcPV2-L1: + (32–37 Cq), ++ (26–31 Cq), +++ (20–25 Cq), and ++++ (14–19 Cq). Samples were considered negative with Cq> 38. All of the cases were then classified into EcPV-2^high^ when viral load was +++ or ++++ and into EcPV-2^low^ when viral load was ++, +, or −.

### 2.4. RNA Extraction and Evaluation of Gene Expression

Five sections (5 µm) were obtained from FFPE samples for RNA extraction, which was performed while using RecoverAll™ Total Nucleic Acid Isolation Kit for FFPE (Invitrogen™) in according to manufacturer’s instructions. EcPV-2-L1 gene expression was tested using specific primers set, probe (Table 2) and Real Time PCR protocols previously described [26]. Briefly, the RNA concentration was evaluated by NanoDrop 2000 (Thermo Fisher Scientific, Waltham, MA, USA) spectrophotometer and 500 ng of RNA were added to the reaction mix for cDNA synthesis. Reverse transcription step (RT) was performed using SuperScript™ IV VILO™ Master Mix (Invitrogen, ThermoFisher Scientific, Waltham, MA, USA) in agreement with the manufacturer’s instructions. To evaluate EcPV2-L1 gene expression, 5 μL of 1:10 diluted cDNA were added to 20 μL PCR mixture at the final concentration of 1×master mix (iTaq Universal Probs Supermix, Bio-Rad, Irvine, CA, USA) 200 nM of probe, 100 nM of each primer combination with the following thermal profile: 95 °C for 10 min., then 39 cycles of 95 °C for 15 s and 60 °C for 60 s in a CFX96™ Real-Time System. RNA was used as the control to exclude possible contaminations by EcPV2 genomic DNA.

Nuclease-free water was used as a negative control. Each sample was tested in triplicates and fluorescence data were collected at the end of the second step of each cycle. In this study, we selected interferon-gamma (*IFNG*)*,* interleukins (*IL)-2, IL5, IL8, IL10, IL12/p35, IL12/p40,* and Transforming growth factor beta 1 (*TGFB1*) to investigate their gene expression. Tests were assessed by RT-qPCR using previously primers tested [28,29,30,31] and summarized in Table 3. Sybr Green Real-Time PCR amplification were performed in a CFX96™ Real-Time System with 5 μL of 1:10 diluted cDNA added to 15 μL PCR mixture at final concentration of 1× master mix (Power SYBR™ Green PCR Master Mix, Applied Biosystems, Thermo Fisher Scientific, Waltham, MA, USA) and 200 nM of each primer combination. The following thermal profile was applied: 95 °C for 10 min., then 50 cycles of 95 °C for 15 s and 60 °C for 30 s. Each sample was tested in triplicates, fluorescence data were collected at the end of the second step of each cycle and, following cycling, the melting curve was determined in the range of 58–95 °C with an increment of 0.01 °C/s, as previously described [32]. Samples were considered to be negative with Cq > 48 in order to evaluate the basal level of expression. Data are expressed as mean Cq of three replicates ± 1 standard deviation. Data related to *IL2* and *IFNG* are expressed as 2^-ΔΔCq^ where ΔCq = Cq (target gene) − Cq (reference gene, *B2M*) and ΔΔCq = ΔCq (samples *TGFB*+) − ΔCq (samples *TGFB*-). The values are given as mean of three replicates ± 1 standard deviation.

### 2.5. Statistical Analysis

Normality was assessed with a Shapiro–Wilk test for all continuous variables. Descriptive statistics were used to describe data and values and they are expressed as medians (Mdn) and interquartile range (IQR). Parametric and non-parametric tests were used to test the hypothesis. The Mann–Whitney U test was performed to assess differences among groups (P/CIS vs SCC). The Wilcoxon Signed rank test was used to compare intra/peritumoral and extratumoral positive cells. T-test was performed in order to evaluate difference in *IL2* and *IFNG* gene expression among groups (*TGFB1*+ vs. *TGFB1*−). Correlation analysis was performed while using the Spearman’s test (ρ). Descriptive statistics were performed using Microsoft Excel; other statistical tests were performed with IBM SPSS (version 21).

## 3. Results

### 3.1. Case Selection, Histological Characterization of Tumors and of the Immune Infiltrate and Ki-67 Index

From our archives were retrospectively selected 20 cases of equine penile epithelial neoplastic lesions; one case was diagnosed as papilloma, two cases as carcinoma in situ (CIS), whereas the 17 remaining ones were confirmed as invasive SCCs. Extratumoral (clean surgical margins) control tissues were available in 10/20 cases. The median age at the moment of the diagnosis was 19 years (IQR = 16–23.5) (mean = 18.73 ± 7.99). Table 4 summarizes all of the histopathological features, including the results of the semi-quantitative evaluation of the immune infiltrate. The median mitotic count was 37.89 (IQR = 22.11–53.69) and median Ki-67 index 31.93 (IQR = 16.58–46.57).

### 3.2. Immunohistochemical Characterization of the Intra/Peritumoral and Extratumoral Immune Infiltrate

CD3^+^lymphocytes were more numerous within intra/peritumoral areas (Mdn = 42.15; IQR = 29.5–82.52) when compared with extratumoral marginal tissues (Mdn = 14.85; IQR = 9.8–25.55) (*p* = 0.013). In numerous samples, the presence of scattered or small clusters of 2–3 CD3^+^cells was observed within tumor lobules, among neoplastic cells. 

FoxP3^+^cells were also statistically increased in intra/peritumoral tissues (Mdn = 25.9; IQR = 12.95–33.15) when compared to extratumoral areas (Mdn = 1.50; IQR = 0.95–3.00) (*p* = 0.011). Numerous FoxP3^+^cells were observed within neoplastic lobules and trabeculae. FoxP3 immunolabeling could be evaluated on 17 cases and nine controls due to scant residual tissue caused by serial recuts of the paraffin block. 

CD20^+^cells were observed both in association with the tumor and in extratumoral tissues and no statistically significant difference in the number of intra/peritumoral (Mdn = 3.00; IQR = 1.25–9.35) and extratumoral (Mdn = 15.75; IQR = 3.17–31.00) tissue was observed (*p* = 0.169). Massive extratumoral infiltration of CD20^+^lymphocytes was frequently observed in the superficial fascia, often arranged in pseudofollicular aggregates (MALT) (Figure 1, Extratumoral CD20). Occasionally, aggregates of CD20^+^lymphocytes were also observed in deeper portions of the penis, among vascular sinuses. 

MUM1^+^cells were more numerous within the intra/peritumoral stroma (Mdn = 48.55, IQR = 23.1–96.12) than in the extratumoral areas (Mdn = 12.55; IQR = 4.67–24.90; *p* = 0.017). The MUM1^+^cells were morphologically classified as plasma cells and they were massively infiltrating tumoral stroma, but were rarely found within neoplastic lobules. In extratumoral tissue, the MUM1^+^cells were arranged as scattered perivascular elements and they were only seldom observed within follicular structures of MALT, where CD20^+^lymphocytes predominated. An increased number of MUM1^+^cells was seen in the superficial fascia, under areas of severe hyperplasia/dysplasia of the epithelium. 

MAC387 was expressed in both macrophages and in neutrophils. The immunolabeling was granular and cytoplasmic. Both cellular populations were more numerous in intra/peritumoral areas (macrophages Mdn = 10.87; IQR = 3.48–15.58; neutrophils Mdn = 15.38; IQR = 3.63–27.96), when compared to extratumoral control tissues (macrophages Mdn = 0.88; IQR = 0.32–3.13; neutrophils Mdn = 0.46; IQR = 0.23–4.28) (*p* = 0.005 and *p* = 0.007). 

IBA1 immunolabeling was observed in cells that were interpreted as histiocytes/macrophages. When compared to the extratumoral tissues (Mdn = 27.75; IQR = 18.30–43.25), IBA1^+^cells were significantly higher in number within the tumoral stroma (Mdn = 7.75; IQR = 3.82–12.57) (*p* = 0.037). These cells were frequently present within the tumoral stroma and also within lobules of neoplastic cells. Examples of the intra/peritumoral and extratumoral markers’ expression are pictured in Figure 1. Figure 2 presents box plots.

### 3.3. Relationship between Immunohistochemical Expression of Markers of Immune Infiltrate, Histopathological Diagnosis and Features, and EcPV-2 Viral Load

Intra/peritumoral CD3 and IBA1 were both increased in SCCs (*p* = 0.039 and *p* = 0.005, respectively) as compared to the CIS/P group, whereas the other variables did not show association with histological diagnosis. None of the histological features was associated with any of the immunohistochemical markers, except intra/peritumoral CD20^+^cells, that were associated with the histological degree of inflammation (*p* = 0.008), being more numerous in tumors with inflammation grade 2 or 3. No differences in terms of mitotic count or Ki-67 index were observed among the EcPV-2^high^ and EcPV-2^low^ groups.

MAC387^+^neutrophils were increased in the EcPV-2^high^ group (*p* = 0.04). No association among histological variables (degree of differentiation, ulceration, vascular invasion, intratumoral necrosis, and quantity of tumoral stroma) and viral load was observed.

Spearman’s correlation (Table 5) indicated a very strong correlation between CD20^+^ and IBA1^+^cells within the tumor environment (ρ = 0.742; *p* < 0.01). Strong positive correlation was also observed between the CD20^+^and MUM1^+^cells (ρ = 0.692; *p* < 0.01), MUM1^+^ and IBA1^+^cells (ρ = 0.606; *p* < 0.01), FoxP3 and MUM1^+^cells (ρ = 0.598; *p* < 0.05) and FoxP3^+^cells and MAC387^+^neutrophils (ρ = 0.564; *p* < 0.01).

### 3.4. EcPV-2 Detection and EcPV-2-L1 Gene Expression

The B-2-microglobulin (*B2M)* gene was amplified in all of the samples, which were therefore considered to be suitable for the investigation of the viral gene *L1*. 18 out 20 (90%) cases were positive for EcPV2-L1 virus DNA (Table 6). The two negative samples were both SCCs. Among the positive samples, 25% showed a mean Cq between 32-37 (+); 10% between 26-31 Cq (++); 20% of tested CCS showed a mean Cq 20–25 (+++); and, 35% between 14–19 Cq (++++). *L1* gene was expressed in 65% of samples (13/20) (Table 3).

### 3.5. Relationship between Immunohistochemical Markers of Immune Infiltrate and RT-qPCR Expression of Cytokines

Concerning cytokines gene expression, interleukin (*IL)10* was expressed in 20% of our samples (4/20), *IL5* on 25% (5/20), *IL12/p35* in 40% (8/20), transforming growth factor beta 1 (*TGFB1)* in 65% (13/20), *IL12/p40* in 85% (17/20), I*L8* in 80% (16/20), *IL2* in 95% (19/20), and interferon gamma (*IFNG)* in 100% (20/20) (Table 6).

The expression of *IL12/p35* was associated with the viral load, being absent in EcPV-2^high^ samples (*p* = 0.007). Samples with normal (+++) or high viral load (++++) showed the expression of *L1* and *IL12/p40* (9 out of 11; Table 6). The 65% of samples were positive for *TGFB1* gene expression; in this group, a significant decrease of the *IFNG* (Mdn= −3.6; IQR=−4.5–2.25) and *IL2* gene (Mdn = 4.00; IQR=2.05-5.45) expression was highlighted (*p* = 0.031 and 0.029, respectively), when compared to *TGFB1*-negative samples (Mdn = −5.1; IQR = −5.7–3.8 and Mdn = 1.55; IQR = 0.575–2.775, respectively) (Figure 3A,B).

### 3.6. Relationship between Immunohistochemical Markers of Immune Infiltrate and rt-qPCR Expression of Cytokines

None of the immunohistochemical markers were associated with *IL5*, *IL8*, *IL12*/*p35*, and *IL12*/*p40* gene expression. The CD20^+^ and IBA1^+^cells were statistically increased in *IL10*-expressing specimens (*p* = 0.022 and *p* = 0.016, respectively), whereas CD3^+^cells were bordering statistical significance in *TGFB1*-positive tumors. It was not possible to test associations of immunohistochemical markers with *IFNG* and *IL2* due to the positivity of these cytokines in all cases (or in all cases, except one for *IL2*), which did not allow stratification.

## 4. Discussion

Recent studies indicate that approximately 15–20% of cancers are caused by viral agents and around 5% by PVs [21,33]. In humans, a significant number of cervical, oropharyngeal, penile, anal, vaginal, and vulvar cancers are induced by mucosal hrHPVs. In horses, PVs infections have been associated with penile, vulvar, vaginal, oropharyngeal [34] and, more recently, with gastric SCC [26,35]. In both these species, tumorigenesis is not an immediate consequence of PVs infection; indeed PVs-induced carcinogenesis can occur after decades of infection [36,37]. It is now accepted that an effective immune control is required in order to prevent PVs persistent infection [8,21] and that the local immune TIME plays a pivotal role during cancer progression [21,38]. During PV infections, the role of the immune system has been shown to change in a stage-dependent manner. In the first stage of infection, anti-viral immunity predominates and the virus has to adopt immune escape strategies to establish persistent infection. Subsequently, PVs-transformed cells reprogram local immune microenvironment and establish a chronic stromal inflammation, which can lead to tumor progression [21].

This study aims at the characterization of the immune infiltrate in equine penile SCCs, and at the assessment of possible associations with histopathological findings and EcPV-2 viral load.

In our study group, 17 cases were diagnosed as epSCCs, 2 as CIS, and one as papilloma. The median age at the moment of the histological diagnosis was 19 years, similarly to previous reports in horses [19], confirming that, as in men, penile SCCs usually develop in advanced age. This finding has been hypothesized to be associated with an age-related reduction of the level of immune surveillance, which could favor a more extensive viral gene expression and appearance or reappearance of PV-induced lesions [8]. Additionally, the papilloma included in this study and one of the two CIS were diagnosed in horses of five and seven years, respectively, suggesting that precursor lesions should be carefully checked in younger animals [17]. 

In our study, 90% of the tumors (18 out of 20) were positive for EcPV-2 DNA, similarly to what has been reported in previous studies [37,39]. In contrast, Arthurs et al., detected less than 50% EcPV-2 positivity in penile tumors [18]. These differences could be due to different detection methods since, in formalin-fixed and paraffin-embedded (FFPE) samples, DNA fragmentation can lead to false negative results when primers set target amplicons that are larger than 200 bp. Both of our samples negative for L1 DNA were histologically diagnosed as SCCs. As in men, equine penile SCCs can also be caused by other agents, such as UV radiation or chronic inflammation [18,40].

Despite a positivity of EcPV-2 DNA in 90% of our samples, only 65% of tumors showed the expression of the *L1* gene. These findings could be due to different factors. The presence of viral DNA without viral gene expression could be associated with a contamination by EcPV-2, which could be non-participating in epSCC pathogenesis and progression or could be due to RNA degradation in FFPE samples. All of the samples that showed *L1* gene expression were associated with a high viral load. As reported in a recent study, viral load could be associated with the progression of lesions with intraepithelial dysplasia usually being associated with a lower HPV16 viral load, when compared to cervical cancer [41]. We could hypothesize that, also in horses, tumor progression might be associated with a higher viral load. Nevertheless, this hypothesis must be tested in a prospective study.

In our study, we specifically investigated the presence of EcPV-2, which is reported to be the most common equine PV in epSCCs [39]. At now, only 13 equine PVs have been described and a recent study demonstrated the presence of EcPV-9 in horse semen of a stallion with penile lesions [42]. Therefore, we cannot exclude that, as for men, other still unknown EcPVs viral types could be involved in epSCCs pathogenesis.

The presence of a statistically significant increased number of different inflammatory cell populations indicate a strong TIME in most of epSCCs, with a marked presence of TILs (tumor-infiltrating lymphocytes), TAMs (tumor-associated macrophages), and TANs (tumor-associated neutrophils). Although mucosal sites usually have an abundant resident immune population, the statistically significant difference between intra/peritumoral inflammation and extratumoral response would suggest a tumor-related or, at least, a tumor-associated effect. Overall, the immune infiltration in epSCC is mainly characterized by MUM1^+^plasmacells and CD3^+^T lymphocytes, as demonstrated by the median of the positive cells/HPF. On the contrary, CD20^+^ B lymphocytes were less represented within tumors’ microenvironment. The findings are similar to what has been reported in humans, where CD3-rich infiltrate was more common than a CD20-rich infiltrate [24]. Similar results have also been previously reported in horses, where CD79^+^lymphocytes were less numerous than CD3^+^cells [25]. 

MUM1 (also known as IRF4), a member of the IRF family, is expressed mainly in the nucleus of B cells, particularly during the final phase of differentiation into plasma cells [43]. Recent investigations have demonstrated that elevated numbers of plasma cells could be associated with worse prognosis [44]. In a study on prostatic cancer, it was postulated that immunosuppressive IgA+ plasma cells within tumors induced CD8+ cell exhaustion and suppressed anti-tumor cytotoxic T cell responses through PD-L1 and IL-10 [45]. Therefore, the presence of immunosuppressive subsets of B cells (Breg) within the TIME to better identify can be hypothesized [46]. The presence of a high number of MUM+ cells in the tumors of our case series could suggest an activation of an immunosuppressive environment, if these cells would be demonstrated to have properties of Breg cells.

FoxP3 is a transcription factor that is currently recognized as the most specific marker for Tregs [47]. FoxP3^+^cells in our study were significantly increased in intra/peritumoral tissues when compared to extratumoral areas. In humans, a higher positivity for FoxP3 has been recognized as an independent predictor of recurrence in penile SCCs [24]. Moreover, FoxP3 expression has also been associated with a poor outcome in other neoplastic conditions, both in humans and animals [48,49]. With this being a retrospective study, follow-up data were not retrieved; a prospective investigation is currently running to establish a possible prognostic role of these cells in epSCCs. 

In this preliminary study, we also observed an increased number of TAMs (MAC387 and IBA1). IBA1 (also known as allograft inflammatory factor, AIF-1) is a protein that is involved in chronic inflammation, and it is expressed by microglia and macrophages. Its role is still not completely understood [50], but in veterinary medicine it has been used as a specific marker for macrophages/microglia, since there are not many available antibodies [51]. In our study, an increased number of IBA1^+^cells was associated with the expression of *IL10*. This finding could be due to an M2-polarization of TAMs within a group of epSCCs, which are able to induce a strong immunosuppressive TIME. Unfortunately, studies on the characterization of equine macrophages are still few [52] and, to the best of the authors’ knowledge, no investigations have been conducted on TAMs in equine tumors. We could also speculate that TGF-β, expressed in most of our samples, could be involved in M2-polarization, as reported recently in hrHPVs infections [53]. *IL10* expression positively correlated with the number of intra/peritumoral CD20^+^cells. With IL10 being recognized as a marker of Bregs in humans, we cannot exclude that an immunosuppressive environment is indeed selected in epSCCs [54].

As for gene expression, *IL12/p35* was differentially expressed in EcPV^high^ and EcPV^low^ groups, being lower or absent in EcPV^high^ tumors. IL-12 is part of IL-12 cytokine family that includes also IL-23, IL-27 and IL-35 [55]. IL-12 and IL-23 share the same IL12p40 subunit; IL-12 is composed of IL12/p40 and IL12/p35, whereas IL-23 by IL12/p40 and IL23/p19. IL-12 is produced by macrophages, B-lymphoblastoid, and dendritic cells, and possesses many functions that are pivotal in innate and acquired immunity. In particular, it is known that IL-12 promotes Th0 differentiation into Th1 and IL2R and IFN-γ expression, bearing an anticancer activity [55]. We postulate that, in our cases, a Th1 activation, as suggested by the expression of *IL-12*, may control viral replication. The absence of association between viral load and *IL12/p40* is most likely due to the presence of IL-23. This cytokine, together with other Th17-related cytokines, are currently under investigation by our group to investigate the possibility of a Th17 activation [56,57]. Because Th17 response is opposed to Th1 and promotes the expression of proinflammatory cytokines like IL-1β, tumor necrosis factor alpha (TNF-α) and IL-8 by macrophages, an imbalance between Th1/Th2 and Th17/Treg in PV-associated lesions could support tumor development in a subgroup of our samples, as reported in cervical cancer [58]. In the present study, the observation of an association between EcPV-2^high^ viral load and a higher infiltration of MAC387^+^ neutrophils would support the hypothesis of a Th17-driven neutrophil activation. 

The results from RT-qPCR indicate the expression of *TGFB1* in most of our samples. The source of TGF- β could be searched in the massive TAMs infiltration (MAC387^+^ and IBA1^+^) that we observed both among tumor cells and in tumor-associated stroma and that has been recognized as driver of immunosuppression-related tumor progression. When grouping samples according to *TGFB1* expression, we observed an increased expression of *IL2* and *IFNG* in samples negative for *TGFB1*- samples and vice versa in *TGFB1*^+^ cases. This is consistent with TGF-β’s known downregulation action on IFN-γ and IL-2.

The limits of this study are the low number of cases that we investigated, together with the use of FFPE samples, which can impact the quality of RNA for RT-qPCR. Moreover, a complete follow-up of the animals, including overall survival and SCC-related death would be helpful in the identification of possible prognostic factors among the immune infiltrate markers. 

## 5. Conclusions

Taken together, our results describe a complex inflammatory environment within epSCCs, with a marked increase of inflammatory populations within the TIME, often characterized by the expression of immune suppressive markers. Our findings also suggest that a Th1-activated TIME might be effective in controlling EcPV-2 viral replication. Further investigation on TIME in epSCCs could identify prognostic factors and, in the future, targets for immunotherapy strategies. 

Prospective studies are currently running in order to overcome these limits and to confirm our preliminary findings. These studies, in the era of the ‘One Health’ approach, could confirm epSCCs as a valid preclinical model for human disease.

## Figures and Tables

**Figure 1 cells-09-02364-f001:**
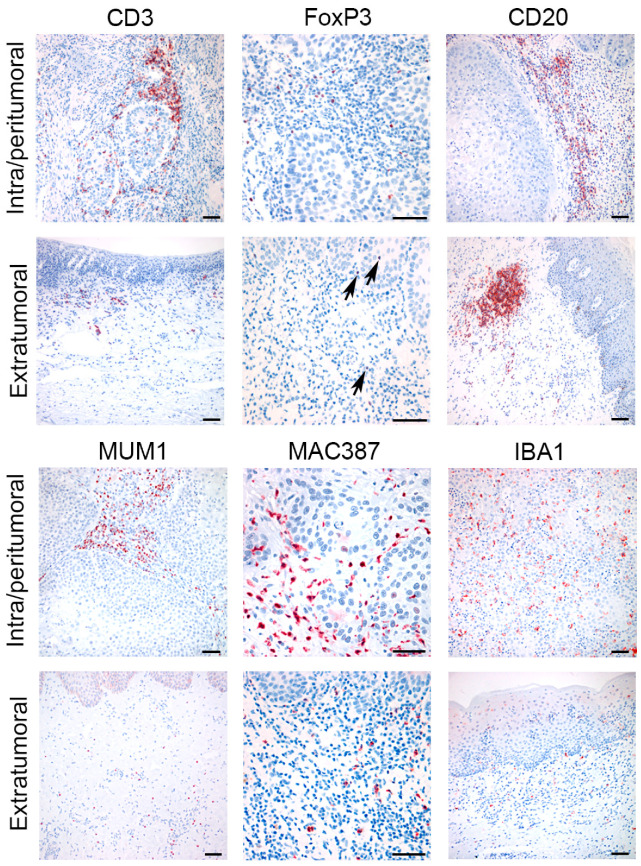
Immunohistochemical expression of different immunohistochemical markers for immune cell population in intra/peritumoral areas of epSCCs and in the extratumoral tissues. CD3 cells were arranged in aggregates within the tumor stroma and, occasionally, also within tumor lobules. FoxP3 cells were numerous within the tumor and scattered in extratumoral tissues (arrows). CD20 could be observed in extratumoral tissues as follicular aggregates (MALT). MUM1, MAC387, and IBA1 cells were more numerous within the tumoral stroma, when compared to the extratumoral tissues. Scale bars: 200 microns.

**Figure 2 cells-09-02364-f002:**
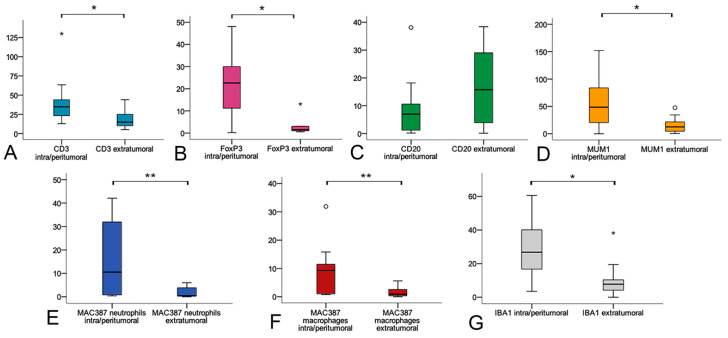
Box plots of the number of CD3 (**A**), FoxP3 (**B**), CD20 (**C**), MUM1 (**D**), MAC387 neutrophils (**E**), MAC387 macrophages (**F**), and IBA1 (**G**) positive cells in the intra/peritumoral areas and in the extratumoral areas. * *p* < 0.05, ** *p* < 0.01 (Mann–Whitney tests).

**Figure 3 cells-09-02364-f003:**
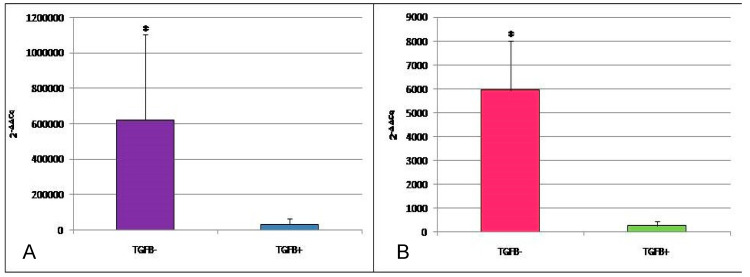
**A**: IFNG expression in equine penile tumor samples. **B**: *IL2* expression in equine penile tumor samples. Data are expressed as 2^−ΔΔCq^ where ΔCq = Cq (target gene) − Cq (reference gene); values are the mean of three test replicates ± 1 standard deviation and ΔΔCq = ΔCq (samples TGFB+) − ΔCq (samples TGFB−). Negative samples were given a Cq 48 fictitious value. * *p* < 0.05.

**Table 1 cells-09-02364-t001:** Antibodies and protocols used for the characterization of the immune infiltrate in equine penile squamous cell carcinomas (epSCCs).

Antibody	Source	Working Dilution; Incubation	Antigen Retrieval	Reaction Pattern
**CD3**	Rabbit polyclonal, Dako	1:200; 1h RT	HIER, ph 9.0, TRIS-Edta	Membrane
**CD20**	Rabbit polyclonal, Dako	1:300; 1h RT	No AR	Membrane
**FOXP3**	Rat monoclonal, eBioscience	1:50; ON 4 °C	HIER, ph 9.0, TRIS-Edta	Nuclear
**MUM1**	Mouse monoclonal, Dako	1:200; 2h RT	HIER, ph 9.0, TRIS-Edta	Nuclear and cytoplasmic
**IBA1**	Goat polyclonal, LSBio	1:50; 2h RT	HIER, ph 9.0, TRIS-Edta	Cytoplasmic
**MAC387**	Mouse monoclonal, Dako	1:100; 2h RT	HIER, ph 9.0, TRIS-Edta	Cytoplasmic
**Ki-67**	Mouse monoclonal, Dako	1:50; 2h RT	HIER, ph 9.0, TRIS-Edta	Nuclear

**Table 2 cells-09-02364-t002:** Primers Set and probes used to evaluate EcPV2 presence and expression.

Gene	Sequences	Accession Number	Bp
Ec-PV2-L1	F-5′-TTGTCCAGGAGAGGGGTTAG-3′	NC_012123.1	81
	R-5′-TGCCTTCCTTTTCTTGGTGG-3′		
pEc-PV2-L1	FAM-CGTCCAGCACCTTCGACCACCA-TAMRA	NC_012123.1	22
Ec-B2M DNA detection	F-5′-CTGATGTTCTCCAGGTGTTCC-3′	NM_001082502.3	114
	R-5′-TCAATCTCAGGCGGATGGAA-3′		
Ec-B2M cDNA expression	F-5′-GGCTACTCTCCCTGACTGG-3′	NM_001082502.3	136
	R-5′-TCAATCTCAGGCGGATGGAA-3′		
pEc-B2M	FAM-ACTCACGTCACCCAGCAGAGA-TAMRA	NM_001082502.3	21

**Table 3 cells-09-02364-t003:** Primers set used in this study to evaluate cytokines genes expression.

Gene	Primer Pairs Sequences	mRNA Position	Genomic Position	Amplicon Length	Accession
*B2M*	F-5′-GGCTACTCTCCCTGACTGG-3′	32-50	chr1:145961271-145964672	136	NM_001082502.3
	R-5′-TCAATCTCAGGCGGATGGAA-3′	147-167			
*TGFB1*	F-5′-CGGAATGGCTGTCCTTTGATG-3′	577-597	chr10:12028778-12030603	127	NM_001081849.1
	R-5′-CCCACGCGGAGTGTGT-TAT-3′	685-703			
*IL5*	F-5′-ACCTGATGATTCCTACTCCTGA-3′	145-166	chr14:42325437+42326396	99	NM_001082499.1
	R-5′-CCCCTTGGACAGTTTGATTCT-3′	223-243			
*IL8*	F-5′-CTGGCTGTGGCTCTCTTG-3′	13-30	chr3:63719744-63720834	133	NM_001083951.2
	R-5′-CAGTTTGGGATTGAAAGGTTTG-3′	122-143			
*IL10*	F-5′-TTCAGCAGGGTGAAGACTTTCT-3′	141-162	chr5:2996651-2997849	107	NM_001082490.1
	R-5′-AAGGCTTGGCAACCCAGGTA-3′	228-247			
*IL12/p35*	F-5′-CTGAGGACCGTCAGCAACAC-3′	243-262	chr19:3775251-3777993	147	NM_001082511.2
	R-5′-GTTCGGGGCGAGTTCCAG-3′	372-389			
*IL12/p40*	F-5′-GATCGTGGTGGATGCTGTTC-3′	629-648	chr14:19332153+19333691	132	NM_001082516.1
	R-5′-TCCACCTGCCGAGAATTCTT-3′	741-760			
*IL2*	F-5′-GAAGAAGAACTCAAACCTCTG-3′	237-257	chr2:105984416+105986321	148	NM_001085433.2
	R-5′-TTCCTGTCTCATCATCATATTC-3′	363-384			
*IFNG*	F-5′-GCTGTGTGCGATTTTGGGT-3′	33-51	chr6:84511911-84513199	130	NM_001081949.1
	R-5′-ATCCAGGAAAAGAGGCCCAC-3′	142-161			

**Table 4 cells-09-02364-t004:** Histopathological features of equine penile tumors.

Histopathological Features		Number of Cases	Percentage (%)
**Histological diagnosis**	SCC	17	85.0
	in situ carcinoma	2	10.0
	Papilloma	1	5.0
**Differentiation grade**	Poorly differentiation	3	17.6
	Moderately differentiated	12	70.6
	Well differentiated	2	11.8
**Vascular invasion**	Present	4	23.5
	Absent	16	76.5
**Ulceration**	Present	15	75.0
	Absent	5	25.0
**Necrosis**	Present	6	30.0
	Absent	14	60.0
**Stroma**	Scant	5	25.0
	Moderate	10	50.0
	Severe	5	25.0
**Inflammation**	Mild	4	20.0
	Moderate	13	65.0
	Severe	3	15.0
	Absent	0	0.0
**Lymphocytes**	Mild	6	30.0
	Moderate	6	30.0
	Severe	8	40.0
	Absent	0	0.0
**Plasma cells**	Mild	9	45.0
	Moderate	5	25.0
	Severe	5	25.0
	Absent	1	5.0
**Macrophages**	Mild	14	70.0
	Moderate	6	30.0
	Severe	0	0.0
	Absent	0	0.0
**Neutrophils**	Mild	5	25.0
	Moderate	12	60.0
	Severe	2	10.0
	Absent	1	5.0
**Eosinophils**	Mild	8	40.0
	Moderate	5	25.0
	Severe	1	5.0
	Absent	6	30.0

**Table 5 cells-09-02364-t005:** Correlation analysis (Spearman rank correlation coefficient, ρ). All of the values are referred to intratumoral positive cells. (n) = neutrophils; (m) = macrophages.

	CD3	FoxP3	CD20	MUM1	MAC387 (n)	MAC387 (m)	IBA1	Mitotic Count	Ki-67 Index
**CD3**	1.000	0.321	0.272	0.472 ^*^	0.341	−0.221	0.399	0.335	0.213
**FoxP3**		1.000	0.494 ^*^	0.598 ^*^	0.564 ^*^	0.047	0.411	0.422	0.257
**CD20**			1.000	0.692 ^**^	0.564 ^**^	0.175	0.742 ^**^	0.107	0.218
**MUM1**				1.000	0.490 ^*^	0.039	0.606 ^**^	0.150	0.480
**MAC387 (n)**					1.000	0.198	0.359	0.026	0.314
**MAC387 (m)**						1.000	0.014	0.430	-0.010
**IBA1**							1.000	0.393	-0.028
**Mitotic count**								1.000	-0.172
**KI67 index**									1.000

* *p* < 0.05, ** *p* < 0.01.

**Table 6 cells-09-02364-t006:** Histological diagnosis: SCC: squamous cell carcinoma; CIS: carcinoma in situ; P: papilloma. RT-PCR data for B2M are expressed as + (amplified) or – (not amplified); Indication of the viral load was give indicating the Cq at which the positivity for L1 was detected: − (>48 Cq), + (32–37 Cq), ++ (26–31 Cq), +++ (20–25 Cq), and ++++ (14–19 Cq). All samples were classified into EcPV2 high when viral load was +++ or ++++ and into EcPV2 low when viral load was ++ or +. RT-qPCR data are expressed as mean Cq ± 1-standard deviation of three replicates. Negative sample are given >48 Cq.

ID	Histological Diagnosis	DNA			cDNA						
		B2M	L1	Viral Load	L1	*B2M*	*IL5*	*IL8*	*IL10*	*IL12/p35*	*IL12/p40*
**1**	SCC	+	+++	**High**	33.5 ± 0.2	**33.0 ± 0.2**	**37.4 ± 0.5**	**35.8 ± 1.5**	>48	>48	>48
2	SCC	+	+	Low	>48	**31.2 ± 0.1**	>48	**35.4 ± 0.3**	>48	**36.7 ± 1.5**	**37.2 ± 0.7**
3	CIS	+	+++	**High**	34.3 ± 0.3	**33.2 ± 0.3**	>48	**31.2 ± 1.6**	>48	>48	**35.4 ± 0.2**
4	SCC	+	++++	**High**	35.1 ± 0.6	**35.5 ± 0.4**	>48	>48	**37.8±0.9**	>48	**36.5 ± 0.7**
5	P	+	+	Low	>48	**32.5 ± 0.3**	**39.8 ± 2.3**	>48	>48	**35.0 ± 1.2**	**36.8 ± 0.1**
6	SCC	+	+	Low	>48	**33.4 ± 1.0**	>48	>48	>48	**45.0 ± 2.1**	**42.3 ± 7.7**
7	SCC	+	++++	**High**	35.9 ± 0.7	**34.5 ± 0.3**	>48	>48	>48	>48	**36.8 ± 0.1**
8	SCC	+	++++	**High**	32.3 ± 0.3	**33.0 ± 0.4**	>48	**34.2 ± 0.2**	>48	>48	**36.9 ± 0.1**
9	SCC	+	++++	**High**	34.9 ± 0.9	**33.0 ± 0.5**	>48	**36.0 ± 0.5**	>48	>48	**35.9 ± 0.2**
10	SCC	+	+	Low	>48	**37.9 ± 0.9**	>48	**34.9 ± 0.7**	>48	**41.2 ± 2.5**	**36.5 ± 0.1**
11	SCC	+	++++	**High**	33.0 ± 0.3	**34.9 ± 0.8**	>48	**35.9 ± 0.6**	>48	>48	**36.5 ± 0.5**
12	SCC	+	+	Low	>48	**34.5 ± 0.3**	>48	**37.2 ± 0.7**	>48	**45.0 ± 3.4**	**37.6 ± 1.3**
13	SCC	+	++	Low	36.6 ± 0.1	**34.1 ± 1.3**	>48	**36.7 ± 0.8**	>48	>48	>48
14	SCC	+	++++	**High**	31.3 ± 2.7	**33.6 ± 0.5**	>48	**36.7 ± 0.3**	>48	>48	**36.7 ± 0.6**
15	SCC	+	++++	**High**	35.6 ± 0.1	**34.2 ± 0.2**	>48	**34.2 ± 0.1**	**39.7±4.6**	>48	>48
16	SCC	+	++	Low	37.8 ± 1.5	**27.9 ± 0.1**	**35.0 ± 1.1**	**26.1 ± 0.1**	**35.5 ± 1.7**	**38.2 ± 3.3**	**34.8 ± 0.1**
17	SCC	+	-	-	ND	**29.4 ± 0.2**	**42.6 ± 6.7**	**33.0 ± 0.7**	**36.5 ± 0.9**	**36.5 ± 1.7**	**35.9 ± 0.6**
18	CIS	+	+++	**High**	31.3 ± 2.7	**32.6 ± 0.3**	**37.5 ± 0.9**	**30.5 ± 0.2**	>48	>48	**36.1 ± 0.6**
19	SCC	+	+++	**High**	35.3 ± 0.1	**31.9 ± 0.5**	>48	**33.4 ± 0.5**	**35.3 ± 1.5**	>48	**36.0 ± 0.1**
20	SCC	+	-	-	ND	**32.1 ± 0.9**	>48	**36.5 ± 0.7**	>48	**45.6 ± 2.3**	**35.5 ± 0.3**

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
