# Peer review of "Equine Penile Squamous Cell Carcinomas as a Model for Human Disease: A Preliminary Investigation on Tumor Immune Microenvironment"

_cells, 2020, doi:10.3390/cells9112364_

Round 1
Reviewer 1 Report
The research hypothesis is interesting, despite the fact that the material comes from horses. Similar immunological and genetic tests can also be performed in people with numerous histopathological types of squamous cell carcinoma. Such a study could confirm epSCCs as significant preclinical analyzes and perespective goal of immunotherapy in people with these tumors, write the authors in their conclusions.
The sections "Introduction, Material and Methods" have been developed synthetically. Figure 1 could be arranged in 3 columns and would then be easier to read.
Immunological studies could be extended to cytometric analyzes.
The authors cite the results of other researchers published in the last 10 years, including the latest ones, i.e. those published in 2020, which is very important in experimental studies.
Reviewer 2 Report
Porcellato and coworkers present the manuscript entitled “Equine penile squamous cell carcinomas as a model 2 for human disease: a preliminary investigation on tumor immune microenvironment (TIME)”. This is a very well performed study in equines cancer which deserves publication. Figures are very nice and data it’s very clear and rigorously analyzed. Thus, I have only minor concerns which must’ve fully addressed before potential publication in Cells.
Title
Please delete TIME from title, and maybe in manuscript as the acronym is used only few times.
Results
Authors Indicate in introduction section that “FoxP3 is an independent predictor of recurrence; the expression of PD-L1 by tumor cells, a CD163+ macrophage infiltration, a non-classical HLA class I upregulation and a low stromal CD8+ T cell infiltration have also been associated with nodal metastasis”. Please indicates if the animals studied showed also recurrence and metastasis. If not, please explain and discuss the utility of these metastasis markers in this study. Also, it will be useful to describe the response to therapy and associate accordingly.
In addition, it will be desirable to explain the functions of FOXP3, PDL1, CD3+ lymphocytes, etc in carcinogenesis and how they are related in the tumor microenvironment maintenance, and tumor initiation and progression for non-expert readers. If not, study remained very descriptive, please discuss in the consequences of changes in markers protein expression in tumor cell biology of equines.
In Table 3: Primers set used in this study to evaluate cytokines genes expression, please indicate the nucleotide positions for the primers in each gene.
3.2. Immunohistochemical characterization of the intra/peritumoral and extratumoral immune infiltrate. Please indicate in the images the Massive extratumoral infiltration of CD20+ lymphocytes (MALT).
Figure 1 is not indicated in the results sections. Please clearly indicates the description of each panel and antibodies in text, as well the morphological features for each image with arrows or asterisk. What’s mean the arrows in Fox3p images?
Finally, it will be useful to delineate a working model from data.
